# Social determinants of health on all-cause and cause-specific mortality in US adults with chronic obstructive pulmonary disease: NHANES 2005–2018

Xiaohan Ma[1☉], Sidi Jian[1☉], Encun Hou[2*], Yujia Wei[3*], Sijing Tu[3*]

1 Graduate School, Guangxi University of Chinese Medicine, Nanning, Guangxi Zhuang Autonomous Region, China, 2 Ruikang Hospital, Guangxi University of Chinese Medicine, Nanning, Guangxi Zhuang Autonomous Region, China, 3 School of Public Health and Management, Guangxi University of Chinese Medicine, Nanning, Guangxi Zhuang Autonomous Region, China

☉ These authors contributed equally to this work.
* encunhou123@163.com (EH), 361491282@qq.com (YW), 93733917@qq.com (ST)

## Abstract

### Background

Social determinants of health (SDoH) are nonmedical societal factors that influence health outcomes. However, their relationship with mortality risk in patients with chronic obstructive pulmonary disease (COPD) remains poorly understood. This study aimed to evaluate the association between SDoH and the risk of all-cause and cause-specific mortality in COPD patients.

### Methods

Data from seven National Health and Nutrition Examination Survey (NHANES) cycles (2005–2018) were analyzed. Kaplan-Meier survival curves and multivariate Cox proportional hazards models were used to assess the association between SDoH and mortality outcomes, including all-cause and cause-specific mortality. The sensitive analyses were performed to check the robustness of the main findings. Nonlinear relationships were explored using restricted cubic spline (RCS) analysis, offering greater flexibility than traditional linear models. Subgroup analyses further assessed the consistency and robustness of findings across different demographic and clinical factors, enhancing the depth and reliability of the results.

### Results

Among 1,551 COPD participants, 506 deaths occurred, including 114 from cardio-cerebrovascular diseases and 130 from cancer. Higher SDoH scores were inversely associated with survival outcomes. After adjusting for confounders, individuals with higher SDoH scores had increased risks of all-cause mortality (hazard ratio [HR] =

**Data availability statement:** The datasets presented in this study can be found in online repositories. The names of the repository/ repositories and accession number(s) can be found below: Detailed information about this study can be found at the NHANES online website: https://wwwn.cdc.gov/nchs/nhanes.

**Funding:** The author(s) received no specific funding for this work.

**Competing interests:** The authors have declared that no competing interests exist.

1.199, 95% confidence intervals [CI]:1.136, 1.264), cancer mortality (HR = 1.236, 95% CI: 1.100, 1.388), and cardio-cerebrovascular mortality (HR = 1.143, 95% CI: 1.022,1.277) compared to those with lower SDoH scores. Sensitivity analyses confirmed the positive correlations between SDoH and mortality in the COPD population. Kaplan-Meier analyses also revealed worse survival outcomes for participants with higher SDoH scores. The agreement between survival analyses and statistical modeling underscores the predictive value of SDoH in this population.

## Conclusion

The results of our study indicate a notable positive correlation between SDoH score and the likelihood of mortality from all causes and specific causes in patients with COPD.

---

## 1. Introduction

Chronic obstructive pulmonary disease (COPD) encompasses chronic bronchitis, emphysema, and small airway obstruction. The disease is characterized by incompletely reversible airflow limitation, persistent inflammation, excessive mucus secretion, and damage to the bronchial mucosal epithelium, which form the primary pathological basis of COPD [1]. Among environmental factors, smoking is widely recognized as the leading trigger of COPD [2–4]. Beyond environmental causes, genetics, gender [5,6], airway hyperresponsiveness [7], lung growth and development[4,8],and infection also significantly contribute to its development. Gender differences may influence smoking history as well as exposure to particulate matter in occupational environments. Furthermore, lung growth and development play a crucial role in determining susceptibility to COPD, and this factor appears to be closely linked to genetic predisposition [7,9]. In addition to these contributing factors, inflammatory mechanisms, oxidative stress, and an imbalance in protease-antiprotease activity are also critical in the disease's progression. These mechanisms result in degeneration, necrosis, and squamous metaplasia of bronchial epithelial cells, along with recurrent injury and repair cycles in the airway walls. Over time, these processes culminate in structural remodeling of the airways, scar formation, and the eventual establishment of COPD [10,11].

The projection for 2020 clearly highlights that COPD is expected to become the third leading cause of death on a global scale, a significant rise from its position as the sixth leading cause in 1990. Additionally, it is anticipated to be the fifth leading cause of years lost due to premature mortality or physical handicap, measured as disability-adjusted life years, up from 12th place in 1990 [5]. Patients affected by COPD experience not only severely compromised lung function but also a broad spectrum of associated comorbidities that exacerbate their condition. Among these, cardiac comorbidities are particularly concerning, as they substantially increase the risk of mortality for these individuals [12]. Furthermore, many COPD patients face difficulties in maintaining employment due to their physical disabilities, often leading

to impairment of working ability or early retirement. This, in turn, contributes to significant socioeconomic losses and increased health-related expenditures. For instance, a survey conducted in 2008 involving 8,217 COPD patients, of whom 48% were male, in rural areas of Xuzhou, China, revealed alarming trends. A considerable proportion of these patients, approximately 36%, required hospitalization due to exacerbations of respiratory symptoms. The total indirect economic loss resulting from these hospitalizations was estimated at a staggering 4, 327, 050yuan, equivalent to $US 678,000 [13]. Overall, COPD is a major contributor to early mortality, high death rates, and significant financial burdens on healthcare systems worldwide [5].

Social determinants of health (SDoH) are non-medical factors that significantly influence health outcomes and overall well-being [14]. The Rainbow Inequality Model, proposed by Dahlgren and Whitehead, is a key framework for understanding health inequalities. It emphasizes the role of social, economic, and environmental factors in shaping health, influencing outcomes through various mechanisms [15,16]. In *A Health Map for the Local Human Habitat*, Barton and Grant further expand this framework by introducing a 'health map' that links health determinants to the ecosystems where humans live. This map includes not only individual characteristics and socio-economic factors, but also the wider environmental and cultural context, emphasizing the interactions between these elements [17].

SDoH variables are categorized into the five key domains outlined by Healthy People 2030: economic stability, education access and quality, healthcare access and quality, neighborhood and built environment, and social and community context [18]. Each domain encompasses factors that can either promote or hinder health, depending on how individuals experience them.

Unfavorable SDoH are closely linked to a range of adverse clinical outcomes, including conditions such as cardiovascular disease (CVD), cancer, and premature death, all of which significantly reduce quality of life and life expectancy [19–21]. Among these, socioeconomic disadvantages stand out as having a particularly profound impact on the morbidity and mortality associated with chronic obstructive pulmonary disease COPD. Individuals facing these disadvantages are at increased risk of more severe COPD outcomes, including complications and premature death. [22].

SDoH significantly influence COPD mortality. Economic stability is a key factor, as individuals with lower incomes face barriers to quality healthcare, leading to delays in diagnosis and inadequate disease management [22]. Education is also important, as those with lower educational attainment often struggle with health literacy, hindering their ability to engage in preventive measures [23]. Limited access to healthcare further worsens disease outcomes for many individuals with COPD [24]. Environmental factors, such as air pollution and poor housing conditions, accelerate COPD progression, particularly in disadvantaged communities. Additionally, social support and experiences of discrimination can affect coping mechanisms, influencing disease management [25]. Addressing these interconnected SDoH is essential for reducing COPD mortality and improving outcomes, requiring targeted interventions that address the root causes of health disparities.

Despite the increasing recognition of the role of SDoH in health outcomes, their impact on mortality risk—specifically cardiovascular and cancer mortality—has not been thoroughly examined in COPD patients. This study aims to fill this gap by assessing the association between SDoH and both all-cause and cause-specific mortality, including deaths related to cardiovascular disease and cancer, using the latest data from the National Health and Nutrition Examination Survey (NHANES).

## 2. Methods

### 2.1. Data source and study design

This study employed a retrospective cohort design using data from the NHANES conducted between 2005 and 2018. The study aimed to evaluate the association between SDoH and mortality outcomes in patients with chronic obstructive pulmonary disease COPD. The study population was derived from a nationally representative sample of the U.S. adult population. The geographical scope of the study encompasses all 50 states and the District of Columbia, ensuring a broad representation of the U.S. population. The NHANES survey employs a complex, stratified, multistage sampling design,

which includes geographic stratification to reflect health conditions across different U.S. regions. This design allows for the examination of regional variations in COPD mortality and the impact of SDoH on health outcomes.

The study utilized publicly available, de-identified data from the NHANES, which is managed by the National Center for Health Statistics (NCHS). The NCHS Institutional Review Board approved all NHANES data collection protocols, and written informed consent was obtained from all participants by the NCHS at the time of data collection. No additional ethical approval was required for the secondary analysis conducted in this study [26].

## 2.2. Study population

For this cohort study, we initially considered 70,190 participants from seven NHANES cycles conducted between 2005 and 2018. To ensure a focused and relevant sample, we applied specific exclusion criteria. Participants younger than 20 years were excluded to maintain an adult study population. We also excluded individuals with missing data on COPD status, SDoH scores, or key covariates, as well as those lacking complete follow-up information. After applying these criteria, the final sample included 1,551 eligible participants. The selection process is detailed in **Fig 1**, ensuring the study's findings are based on a well-defined and representative population. This refined cohort provides a robust foundation for analyzing the association between SDoH and mortality outcomes in individuals with COPD.

## 2.3. Definition of SDoH score

Self-reported data on eight sub-items of SDoH across five domains were operationalized based on the criteria outlined in the *U.S. Healthy People 2030* initiative and prior studies [18,19]. Detailed definitions of these domains and sub-items are

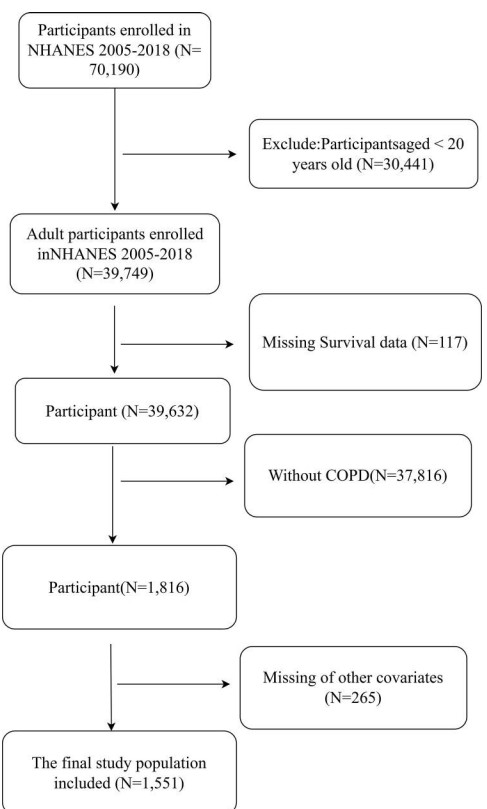

**Fig 1. The flow chart of the included participants in this study.**

provided in S1 Table. The effect of the accumulation of unfavourable SDoH within an individual on mortality was explored by summing eight dichotomous SDoH, each with a value of 0 for each favourable level and 1 for each unfavourable level, to form a cumulative SDoH variable. In accordance with prior research, participants were divided into two groups based on the extent of unfavorable SDoH: those with a lower burden, defined by SDoH scores of 2 or less, and those with a higher burden, indicated by SDoH scores greater than 2 [27,28].

## 2.4. Definition of COPD

COPD was diagnosed based on one or more of the following criteria [29]: (1) a post-bronchodilator FEV1/FVC ratio < 0.70; (2) a diagnosis of emphysema confirmed by a physician or healthcare professional; or (3) for individuals aged 40 years or older, a history of smoking accompanied by chronic bronchitis, and treatment with medications such as selective phosphodiesterase-4 inhibitors, mast cell stabilizers, leukotriene modifiers, or inhaled corticosteroids.

## 2.5. Mortality outcomes

The NCHS provided the Public-Use Linked Mortality Files, which were utilized to determine mortality outcomes in this study (https://www.cdc.gov/nchs/data-linkage/mortality.htm). The primary outcomes assessed were mortality rates for all causes, cardio-cerebrovascular disease (ICD-10 codes I00-I09, I11, I13, I20-I25, I26-I51, and I60-I69), and cancer (ICD-10 codes C00-C97).

## 2.6. Covariables

The variables analyzed in this study included a range of demographic and socioeconomic factors. Age, sex, and ethnicity were recorded, with ethnicity categorized into four groups: Mexican American, Non-Hispanic Black, Non-Hispanic White, and Other. Smoking status was classified based on lifetime cigarette use into three categories: never (fewer than 100 cigarettes), former (more than 100 cigarettes but not currently smoking), and current (more than 100 cigarettes and actively smoking). Alcohol use was classified based on daily and binge drinking frequency. Heavy use was defined as ≥3 drinks per day for women or ≥4 for men, or binge drinking on 5 or more days per month. Moderate use was defined as up to 2 drinks per day for women or 3 for men, or binge drinking at least twice a month, with a separate category for those with a history of daily binge drinking. Body mass index (BMI) (kg/m²) was calculated by dividing weight (kg) by height squared (m²) and categorized into three groups: < 25 kg/m², 25–29.9 kg/m², and >29.9 kg/m² [30]. Hypertension was diagnosed using a combination of factors, including the use of antihypertensive medication, self-reported history of hypertension, and the average of three blood pressure measurements. DM was determined by self-reported diagnosis or current use of antihyperglycemic medications [31]. A diagnosis of CVD was indicated by any affirmative answer to having been informed of congestive heart failure, coronary heart disease, angina, a heart attack, or a stroke [32]. The presence or absence of cancer is based on patient self-report.

## 2.7. Statistical analyses

Sample weighting, clustering, and stratification were systematically applied in all analyses, which was essential to ensure that the results were accurately representative of the national population. Finally, we constructed individual-level sampling weights across the 7 consecutive NHANES survey cycles by using the 2-year mobile examination center weight divided by 7. This method is suitable for multi-stage probability sampling of NHANES. In the baseline analysis, the COPD population was classified according to SDoH levels. Continuous variables were expressed as mean ± standard error (SE), and weighted analysis of variance was used for differences between groups. Categorical variables are expressed as numbers (percentages) and compared using a weighted chi-square test. Kaplan-Meier survival analysis was used to estimate all-cause mortality and cause-specific mortality survival over time and stratified according to SDoH status in the COPD population. Weighted

multivariate Cox proportional hazards regression models were applied to evaluate the association between SDoH and both all-cause and disease-specific mortality in the COPD cohort. Several models were constructed and adjusted to varying degrees: the coarse model, which did not adjust for any covariates; Model 1, which adjusted for age, gender, and ethnicity; and Model 2 (fully adjusted), which further accounted for smoking, alcohol consumption, BMI, diabetes mellitus (DM), hypertension, CVD, and cancer, building on Model 1. To investigate the robustness of the relationship between SDoH and mortality in the COPD population, we performed sensitivity analyses using two different approaches for categorizing SDoH. First, we treated SDoH as a continuous variable, allowing for the examination of linear trends across varying levels of SDoH. Then, we treated SDoH as a categorical variable, dividing it into three groups based on tertile spacing: Q1 (≤2), Q2 (3–4), and Q3 (≥5). This division allowed us to assess the effects of different levels of social determinants on mortality risk, comparing the lowest group (Q1) with higher groups (Q2 and Q3) to better understand the potential impact of increasing SDoH on mortality outcomes within the COPD population. Smooth curve fitting based on the generalised additive model and the Cox model with restricted cubic splines (RCS) is used to further assess potential non-linear relationships between SDoH and all-cause and cause-specific mortality. Stratified analyses were performed according to age, sex, ethnicity, hypertension, CVD and cancer to assess whether these associations were consistent across subgroups and to identify potential influencing moderators. This multifaceted approach ensures a robust and comprehensive assessment of potential factors that may modulate the impact of SDoH on mortality.

All statistical tests were two-sided and P values less than 0.05 were considered statistically significant. All statistical analyses were performed using R and the EmpowerStats software.

## 3. Results

### 3.1. Baseline characteristics

Table 1 presents the baseline characteristics of the study population, categorized by SDoH score quantiles. The weighted mean age of participants was 60.3±0.4 years. Statistical comparisons revealed significant differences between groups for variables such as age, sex, ethnicity, smoking status, alcohol consumption, BMI, and the presence of DM, CVD, or cancer (P<0.05). However, no significant differences were found in the prevalence of hypertension (P>0.05). These findings suggest that demographic and lifestyle factors are associated with variations in SDoH, which may, in turn, influence the overall health status of this population.

### 3.2. Association of SDoH with mortality in COPD

Table 2 demonstrates that in the study population, higher SDoH (continuous) scores were associated with an increased risk of all-cause and cause-specific mortality in COPD patients. This relationship remained consistent across all three models, with statistically significant trends (P<0.05). In fully adjusted Model 2, the hazard ratio (HR) for all-cause mortality among COPD patients with SDoH scores >2 was 2.005 (95% CI: 1.613,2.493) compared to those with SDoH ≤ 2. For cause-specific mortality, the HR for cancer-related mortality was 2.307 (95% CI: 1.423, 3.738), and the HR for cardio-cerebrovascular mortality was 1.377 (95% CI: 0.787.2.409). These results indicate that higher SDoH scores are associated with increased mortality risk, highlighting the potential prognostic significance of SDoH in patients with COPD.

Kaplan-Meier survival analysis further supported these findings, showing significant correlations between SDoH levels and survival probabilities for all-cause mortality, cancer mortality, and cardio−cerebrovascular disease mortality (P<0.05). As shown in Fig 2, participants with higher SDoH scores exhibited lower survival probabilities.

### 3.3. RCS analysis

Smoothing curves further validated the positive linear relationships (Fig 3 A–3C). In a fully adjusted RCS model accounting for confounding factors, higher SDoH scores in COPD patients were associated with an increased risk of all-cause, cancer, and cardio-cerebrovascular mortality.

**Table 1. Baseline characteristics of participants with COPD according to SDoH in NHANES.**

| Variable | Total | >2 | ≤2 | Pvalue |
|---|---|---|---|---|
| Age, mean±SE | 60.3±0.4 | 58.5±0.6 | 61.6±0.5 | < 0.0001 |
| Sex (N,weighted %) | | | | 0.003 |
| Female | 659(48.2) | 411(53.9) | 248(43.9) | |
| Male | 892(51.8) | 451(46.1) | 441(56.1) | |
| Ethnicity (N,weighted %) | | | | < 0.0001 |
| Mexican American | 84(1.9) | 54(2.8) | 30(1.2) | |
| Non-Hispanic Black | 273(7.2) | 175(11.0) | 98(4.2) | |
| Non-Hispanic White | 1000(82.5) | 497(74.9) | 503(88.3) | |
| Other | 194(8.4) | 136(11.3) | 58(6.2) | |
| Smoke (N,weighted %) | | | | < 0.0001 |
| former | 729(46.3) | 333(36.6) | 396(53.8) | |
| never | 242(17.4) | 104(12.2) | 138(21.3) | |
| now | 580(36.3) | 425(51.3) | 155(24.9) | |
| Drink (N,weighted %) | | | | < 0.0001 |
| former | 470(26.6) | 285(32.8) | 185(22.0) | |
| never | 112(6.1) | 73(7.4) | 39(5.1) | |
| now | 969(67.3) | 504(59.8) | 465(73.0) | |
| BMI (N,weighted %) | | | | 0.001 |
| <25 | 438(26.8) | 259(28.8) | 179(25.2) | |
| >29.9 | 614(40.4) | 357(44.4) | 257(37.4) | |
| 25-29.9 | 499(32.8) | 246(26.7) | 253(37.4) | |
| Hypertension (N,weighted %) | | | | 0.1 |
| no | 569(42.3) | 304(38.8) | 265(45.1) | |
| yes | 982(57.7) | 558(61.2) | 424(54.9) | |
| DM (N,weighted %) | | | | 0.004 |
| DM | 355(18.6) | 225(22.5) | 130(15.7) | |
| no | 1196(81.4) | 637(77.5) | 559(84.3) | |
| CVD (N,weighted %) | | | | < 0.0001 |
| no | 1082(74.0) | 561(66.6) | 521(79.6) | |
| yes | 469(26.0) | 301(33.4) | 168(20.4) | |
| Cancer (N,weighted %) | | | | < 0.001 |
| no | 1238(77.2) | 728(83.0) | 510(72.8) | |
| yes | 313(22.8) | 134(17.0) | 179(27.2) | |

Abbreviations: SDoH, social determinants of health; COPD, Chronic Obstructive Pulmonary Disease; BMI, body mass index; DM, diabetes mellitus; CVD, cardiovascular disease.

Data are presented as weighted mean ±SE or weighted frequencies (weighted percentages).

## 3.4. Sensitivity and Subgroup analysis

To verify the stability of the results, we performed sensitivity analyses by analyzing SDoH both as a categorical variable (divided into tertiles). The sensitivity analysis confirmed a positive correlation between SDoH and mortality in COPD patients, consistent across both methods of analysis (S2 Table). **Fig 4A**-4C) presents a stratified analysis of the association between SDoH and mortality (all-cause, cancer, and cardio-cerebrovascular disease). The interaction P-value was not statistically significant (>0.05), suggesting that factors such as age, sex, ethnicity, hypertension, CVD did not significantly impact the relationship between SDoH and all-cause mortality. Similarly, no significant interaction was observed between

**Table 2. Weighted Cox regression analysis of SDoH and long-term mortality in patients with COPD.**

| | Crude model | | Model 1 | | Model 2 | |
|---|---|---|---|---|---|---|
| **All-cause mortality** | HR (95%CI) | P | HR (95%CI) | P | HR (95%CI) | P |
| SDoH (continuous) | 1.155(1.109,1.203) | <0.0001 | 1.280(1.218,1.345) | <0.0001 | 1.199(1.136,1.264) | <0.0001 |
| SDoH (multi-category) | | | | | | |
| ≤2 | ref | | ref | | ref | |
| >2 | 2.041(1.656,2.514) | <0.0001 | 2.534(2.075,3.096) | <0.0001 | 2.005(1.613,2.493) | <0.0001 |
| **Cancer mortality** | | | | | | |
| SDoH (continuous) | 1.168(1.082,1.261) | <0.0001 | 1.293(1.163, 1.437) | <0.0001 | 1.236(1.100, 1.388) | <0.001 |
| SDoH (multi-category) | | | | | | |
| ≤2 | ref | | ref | | ref | |
| >2 | 2.277(1.497,3.464) | <0.001 | 2.846(1.808, 4.480) | <0.0001 | 2.307(1.423, 3.738) | <0.001 |
| **Cardio−cerebrovascular disease mortality** | | | | | | |
| SDoH (continuous) | 1.122(1.024,1.229) | 0.014 | 1.275(1.158,1.403) | <0.0001 | 1.143(1.022, 1.277) | 0.019 |
| SDoH (multi-category) | | | | | | |
| ≤2 | ref | | ref | | ref | |
| >2 | 1.580(1.006,2.483) | 0.047 | 2.112(1.390,3.207) | <0.001 | 1.377(0.787, 2.409) | 0.263 |

Crude model: No adjustment for any potential influence factors.

Model 1: Adjusted for age, sex, ethnicity.

Model 2: Adjusted for age, sex, ethnicity, drinking status, smoke, BMI, DM, hypertension, CVD, cancer.

Abbreviations: CI, confidence interval; HR, hazard ratio. Abbreviations: SDoH, social determinants of health; COPD, Chronic Obstructive Pulmonary Disease;

BMI, body mass index; DM, diabetes mellitus; CVD, cardiovascular disease.

SDoH and cancer mortality in the age, sex, race, hypertension, and cancer subgroups (P for interaction > 0.05), but there was an interaction in the CVD subgroup (P for interaction < 0.05). These findings suggest that demographic and clinical factors, including age, sex, ethnicity, hypertension and cancer, do not alter the strength or direction of the association between SDoH and all-cause and cancer mortality. In addition, in the stratification of SDoH with cardio−cerebrovascular death, we found an interaction at age (P for interaction = 0.003).

## 4. Discussion

To our knowledge, this is the first retrospective study to investigate the relationship between SDoH and mortality in patients with COPD, offering novel insights into how social determinants influence survival outcomes. Our findings demonstrate that higher SDoH scores are significantly associated with increased risks of all-cause, cardio-cerebrovascular disease, and cancer-specific mortality, even after adjusting for key confounders such as age, sex, ethnicity, smoking, drinking habits, BMI, DM, hypertension, CVD, and cancer. In the RCS analysis, the relationship between SDoH and mortality—whether from all causes, cardiovascular disease, or cancer—exhibited a linear positive trend. Sensitivity analyses further confirmed this result. Rigorous cross-validation across multiple analytical models further confirmed the robustness and reliability of these findings.

As observed in our study, SDoH were found to be significantly associated with cancer-specific mortality risk. Cohort studies have demonstrated that vulnerable SDoH contribute to poor physical and mental health, delays in medical and surgical treatments, and an increased risk of both all-cause and cancer-related mortality in patients with cancers such as breast and pancreatic cancer [33–36]. These nonclinical factors, including economic stability, the neighborhood and built environment, access to health and healthcare services, and the social and community context, play a pivotal role in shaping cancer survival rates [37].

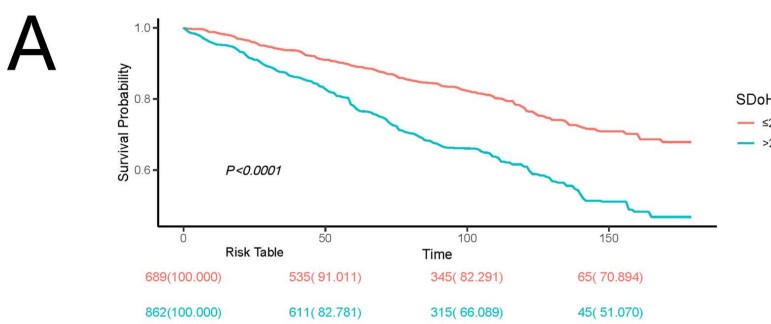

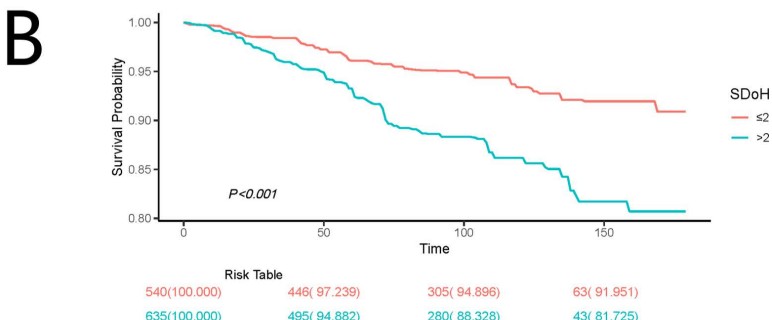

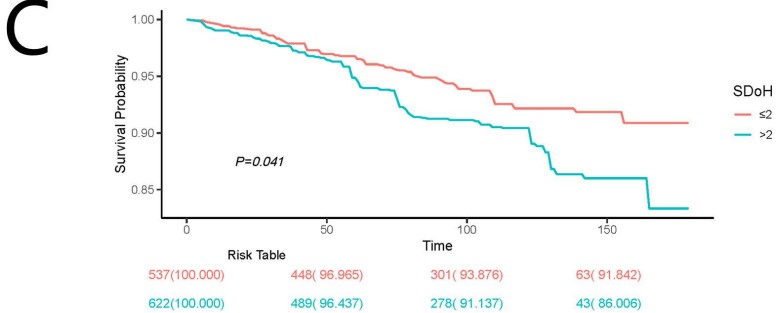

**Fig 2. Kaplan-Meier survival curve of all-cause mortality (A), cancer mortality (B), and cardio −cerebrovascular disease mortality (C) by SDoH.**

For instance, Ross and Wu demonstrated that higher educational attainment indirectly improves health outcomes by fostering a more favorable occupational and economic situation. Their analysis suggested that education is linked to better employment opportunities and increased income, which leads to a more stable and advantageous economic environment [38]. This improvement in socioeconomic status contributes to better outcomes for cancer patients by influencing cancer characteristics and reducing the likelihood of presenting with distant metastases at diagnosis. On the other hand, individuals with lower socioeconomic status face a multitude of intersecting challenges, resulting in significant cancer health inequities [39]. Patients living in socioeconomically disadvantaged areas often encounter substantial barriers to accessing cancer screening, timely diagnosis, and effective treatment services. As a result, they are more likely to present with advanced-stage disease, which negatively impacts their prognosis [40–43]. Several studies analyzing cancer population data have consistently shown that patients from deprived socioeconomic backgrounds experience worse cancer

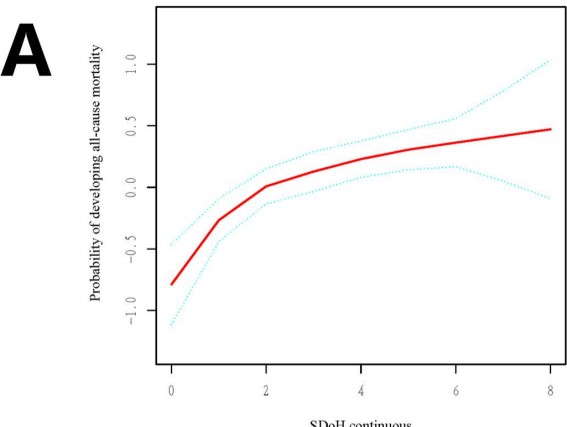

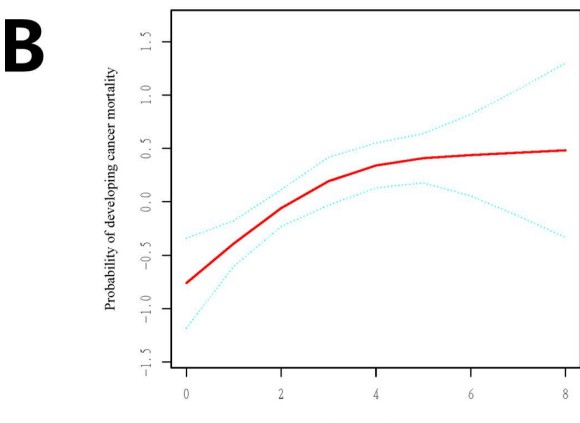

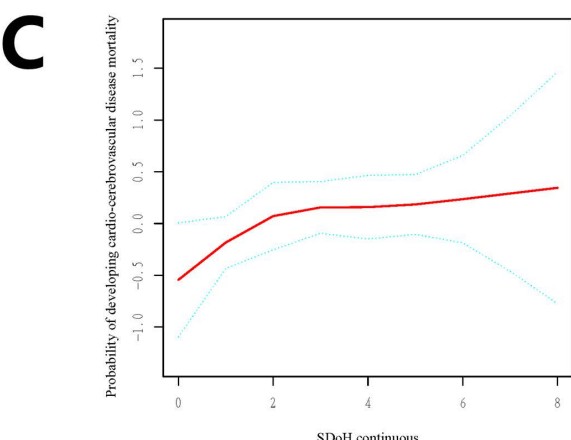

**Fig 3. RCS for the association between SDoH and the probability of all-cause (A), cancer mortality (B) and cardio −cerebrovascular mortality (C) in patients with COPD.**

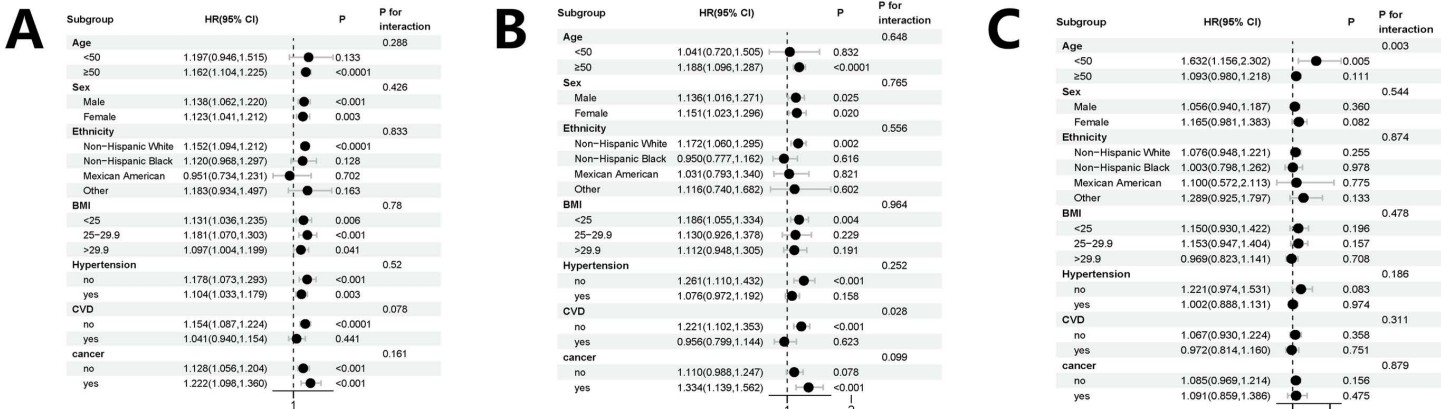

**Fig 4. Stratified analyses of the associations between SDoH and all-cause mortality (A), cancer mortality (B), cardio−cerebrovascular mortality (C).** Adjusted for drinking status, smoke, DM (except for grouping covariates). Abbreviations: CI, confidence interval; HR, hazard ratio. Abbreviations: SDoH, social determinants of health; COPD, Chronic Obstructive Pulmonary Disease; BMI, body mass index; DM, diabetes mellitus; CVD, cardiovascular disease.

outcomes, including higher mortality rates and poorer overall survival [44–46]. These findings highlight the deep and multifaceted relationship between SDoH and cancer risk, emphasizing the urgent need to address these determinants through targeted interventions. Increasing awareness of the profound role that SDoH play in cancer outcomes is essential for developing policies and initiatives aimed at reducing health disparities. By addressing these disparities, healthcare systems can work toward ensuring equitable access to cancer care, ultimately improving survival outcomes for all patients, regardless of their socioeconomic status.

SDoH are strongly associated with an elevated risk of cardiovascular mortality, serving as a key factor in health outcomes across different populations. Our results are consistent with prior research, which has highlighted how unfavorable SDoH contribute to higher rates of premature mortality from all causes [19]. Additionally, two recent studies have shown that a higher burden of adverse SDoH during midlife is linked to an increased risk of both fatal and nonfatal cardiovascular events, such as coronary heart disease and stroke [47,48]. In a similar vein, research examining social determinants of cardiovascular health in U.S. adolescents has found a significant correlation between SDoH and long-term cardiovascular outcomes [49]. Furthermore, a study conducted in China also found that social determinants had strong associations with both all-cause and disease-specific mortality [50]. Lastly, a Danish study suggested that SDoH are linked to poorer prognosis in patients with cardiovascular disease [51].

SDoH encompass a broad, multidimensional set of social, economic, and environmental factors that influence health outcomes throughout a person's life. These determinants affect individuals based on where they are born, raised, live, work, and engage in daily activities. Their impact extends beyond individual health behaviors to shape systemic health disparities, including the development and progression of hypertension and CVD. Specifically, unfavorable SDoH are associated with inadequate housing conditions, poverty, food insecurity, limited access to healthcare, and neighborhood environments that lack green spaces for physical activity. Furthermore, structural inequities, such as systemic racism and economic instability, exacerbate chronic stress, leading to physiological consequences, including increased sympathetic nervous system activity, heightened systemic inflammation, and an elevated risk of hypertension and CVD [52].

Furthermore, limited access to affordable and high-quality healthcare exacerbates these risks, delaying the early diagnosis and timely management of conditions such as hypertension, which in turn accelerates the progression of vascular diseases. As a result, high rates of morbidity and mortality are observed, particularly among socioeconomically disadvantaged groups. This delayed care and inadequate management of chronic conditions contribute to the increased

prevalence of cardiovascular diseases and poor health outcomes [53]. Addressing these complex, interconnected issues is essential to improving cardiovascular health outcomes and reducing health inequities across populations. By tackling these disparities and promoting equitable access to healthcare, we can help reduce the burden of CVD, enhance overall health outcomes, and break the cycle of poverty that exacerbates these conditions

SDoH affect health through a variety of mechanisms, particularly in patients with COPD. Social determinants such as income, education, and social status can indirectly influence health by affecting an individual's health behaviors and access to healthcare. For example, low-income groups may not have access to essential healthcare resources or may face difficulties obtaining adequate nutrition and medical care due to financial constraints, which directly contribute to the deterioration of health [54,55]. Additionally, mental health issues are often associated with lower SDoH scores, which can further exacerbate the burden of chronic diseases such as COPD [56,57]. Poor mental health can impair an individual's ability to manage their condition effectively, leading to worsened physical health outcomes. Thus, the impact of SDoH on health is multifaceted, involving a complex interplay of social, economic, and psychological factors. Addressing these determinants requires a holistic approach to care that considers not only the medical needs of individuals but also the broader socioeconomic and mental health factors that influence health outcomes.

Physical activity has been shown to have a significant effect on improving the health of COPD patients, especially in groups with low SDoH scores. According to relevant studies, regular physical activity not only significantly improves lung function in patients, but is also effective in alleviating mental health problems such as depression and anxiety [58]. These mental health problems are more prevalent in COPD patients with low socioeconomic status, which in turn can negatively affect their disease prognosis [59,60]. Therefore, physical activity interventions may serve as an effective means of mitigating the impact of adverse SDoH, particularly for patients living in low-income and resource-limited settings. By improving both physical and mental health, such interventions can enhance overall well-being and quality of life for these patients, potentially leading to better disease management and outcomes.

In addition to individual-level interventions, community-based aging interventions may also be effective in addressing the health challenges faced by low-SDoH groups. Research has demonstrated that a biobehavioral-environmental approach, such as the "community aging in place, advancing better living for elders" model, can help reduce the burden of chronic diseases and disability among low-income older adults by improving their living environment, increasing social support, and providing accessible healthcare resources [61]. For COPD patients in particular, community-based interventions can assist individuals in better managing their disease by enhancing the availability of medical care, promoting healthy behaviors, and fostering social interactions. These interventions can create an environment that not only addresses the physical health needs of patients but also supports their mental and social well-being, crucial factors for those with chronic conditions.

Our study offers several key strengths. It utilized a large, nationally representative sample from the NHANES cohort, ensuring the findings are generalizable and replicable across broader populations. The prospective design of the study, combined with thorough adjustments for potential confounders, further enhances the reliability of the results and minimizes the risk of bias. These findings underscore the potential impact of policy interventions targeting improvements in SDoH. By addressing these factors, such interventions could help reduce health disparities and improve survival rates among COPD patients, ultimately contributing to a more equitable healthcare system and better health outcomes for affected populations.

## 5. Limitations

The current study has several limitations that warrant acknowledgment. First, its cross-sectional design limits our ability to establish causal relationships between COPD and SDoH. Second, as with many observational studies, residual confounding may exist due to measurement errors or the omission of unmeasured variables. Although we adjusted for numerous covariates, unexplained confounders could introduce bias and potentially affect the accuracy and generalizability of our

findings. Additionally, some important aspects of SDoH, such as experienced racism, discrimination, and social support, were not comprehensively assessed in NHANES, which may have influenced the results. Replication of these findings in larger, prospective cohort studies is therefore necessary. Finally, the results of this study may be most applicable to the U.S. population. Further research is needed to determine whether these findings can be generalized to other populations or settings.

## 6. Conclusions

Our findings indicate a strong positive association between SDoH scores and the risk of both all-cause and cause-specific mortality in patients with COPD. This relationship underscores the critical importance of addressing SDoH to enhance survival and overall health outcomes in this population. Integrating SDoH considerations into clinical practice and public health strategies for COPD management holds promise but requires further validation through larger prospective cohort studies and clinical trials.

## Supporting information

**S1 Table. Definitions of social determinants of health domains and sub-items.**
(DOCX)

**S2 Table. Cox regression analysis of SDoH (by tertile) and long-term mortality in patients with COPD: Sensitivity analyses.**
(DOCX)

## Author contributions

**Formal analysis:** Xiaohan Ma, Sidi Jian, SiJing Tu.

**Investigation:** Xiaohan Ma, Encun Hou, Yujia Wei, SiJing Tu.

**Methodology:** Xiaohan Ma, Sidi Jian, Encun Hou, Yujia Wei, SiJing Tu.

**Project administration:** Yujia Wei.

**Resources:** Xiaohan Ma, Sidi Jian, Yujia Wei.

**Software:** Xiaohan Ma, Sidi Jian, Encun Hou, Yujia Wei, SiJing Tu.

**Supervision:** Sidi Jian, Encun Hou, SiJing Tu.

**Validation:** Sidi Jian, SiJing Tu.

**Visualization:** Xiaohan Ma, SiJing Tu.

**Writing – original draft:** Xiaohan Ma, Sidi Jian, Encun Hou, Yujia Wei, SiJing Tu.

**Writing – review & editing:** Xiaohan Ma, Sidi Jian, Encun Hou, Yujia Wei, SiJing Tu.

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
