## [Decision Letter · Decision Letter 0]

11 Feb 2025

PONE-D-24-59243Social determinants of health on All-Cause and Cause-specific Mortality in US Adults with chronic obstructive pulmonary disease: NHANES 2005–2018.PLOS ONE

Dear Dr. Tu,

Thank you for submitting your manuscript to PLOS ONE. After careful consideration, we feel that it has merit but does not fully meet PLOS ONE’s publication criteria as it currently stands. Therefore, we invite you to submit a revised version of the manuscript that addresses the points raised during the review process.

 The manuscript is scientifically sound and present several robust methodological elements. Yet, after a thorough review, our reviewers have provided feedback highlighting areas that require revision, namely in the analysis and discussion of results, to me more targeted and clear. In light of their comments, we kindly request that you undertake a major revision of your manuscript to address the concerns raised. Please carefully consider each suggestion and provide a detailed response explaining the changes made.

We look forward to receiving your revised manuscript.

Kind regards,

Angela Mendes Freitas

Academic Editor

PLOS ONE

https://bmcpsychiatry.biomedcentral.com/articles/10.1186/s12888-024-06159-3

In your revision ensure you cite all your sources (including your own works), and quote or rephrase any duplicated text outside the methods section. Further consideration is dependent on these concerns being addressed.

Reviewers' comments:

Reviewer's Responses to Questions

**Comments to the Author**

1. Is the manuscript technically sound, and do the data support the conclusions?

Reviewer #1: Yes

Reviewer #2: Yes

2. Has the statistical analysis been performed appropriately and rigorously? 

Reviewer #1: Yes

Reviewer #2: Yes

3. Have the authors made all data underlying the findings in their manuscript fully available?

Reviewer #1: Yes

Reviewer #2: Yes

4. Is the manuscript presented in an intelligible fashion and written in standard English?

Reviewer #1: Yes

Reviewer #2: No

5. Review Comments to the Author

Reviewer #1: Dear authors,

I have finished review of this research paper examining the relationship between social determinants of health (SDoH) and mortality outcomes in COPD patients.

The study demonstrates several robust methodological elements. The researchers appropriately utilized data from the National Health and Nutrition Examination Survey (NHANES), a well-established and nationally representative database. Their statistical approach, incorporating both Kaplan-Meier survival curves and multivariate Cox proportional hazards models, provides complementary perspectives on mortality risk. The inclusion of restricted cubic spline analysis to examine potential non-linear relationships shows sophisticated analytical thinking.

However, there are some notable limitations should be revised before its publication:

1. Data Analysis Concerns:

While the statistical analysis is generally sound, several issues warrant attention. The researchers dichotomized the SDoH scores using a cutoff of 2, but the rationale for this specific threshold isn't thoroughly explained. This artificial dichotomization might oversimplify the complex nature of social determinants. Additionally, the study could have benefited from sensitivity analyses using different SDoH score cutpoints to test the robustness of their findings.

2. Interpretation and Generalizability:

The authors appropriately acknowledge that their findings may be most applicable to the U.S. population, given the NHANES data source. However, they could have provided more discussion about how their results compare to similar studies in other countries or healthcare systems. The linear relationship found between SDoH and mortality outcomes seems somewhat surprising given the complex nature of social determinants, and this finding deserves more critical examination.

3. Discussions:

A more detailed discussion of the potential mechanisms linking SDoH to mortality outcomes, or analysis of potential interactions between different SDoH components are recommended. For example, consider discussing Physical activity and prevention of mental health complications: an umbrella review. This would provide valuable insights into how physical activity interventions could potentially mitigate some of the adverse effects of poor SDoH in COPD patients, particularly given that the authors described increased mortality risks associated with higher SDoH scores. Also, The effectiveness of community ageing in place, advancing better living for elders as a biobehavioural environmental approach for disability among low-income older adults is also recommended. This article would be relevant for understanding community-based interventions that could help address SDoH factors in vulnerable populations, similar to those identified in your COPD study.

Thank you for your valuable contributions to our field of research. I look forward to receiving the revised manuscript.

Reviewer #2: The authors of "Social determinants of health and chronic obstructive pulmonary disease" present a wide range of results and methods applied. The text is generally organized and it is scientifically sound. It is, nonetheless, easy to get lot in the variety of results presented and discussed as the text do not focus only in COPD and also addresses other health outcomes. This drives the reader away from the main topic.

I recommend a revision in the sense of clarifying and refocusing the discussion, maybe organize it in sections

A couple of minor suggestions:

Introduction:

The authors do a suitable presentation of the health determinants framework (although missing key references such as Dahlgren and whitehead work or Barton). Nonetheless, I consider that the text needs to be arranged in a way that the SDoH are presented regarding its impact on COPD, or respiratory mortality, instead of generally.

I also recommend authors to include the aim of the text in a clearer way (and more aligned with the redaction present at the abstract)

Methods:

I suggest including a study design section to provide an general overview of the steps that will follow

The geographical amplitude of the study population is missing (e.i. where were those persons living )

6. PLOS authors have the option to publish the peer review history of their article (what does this mean? ). If published, this will include your full peer review and any attached files.

**Do you want your identity to be public for this peer review?** For information about this choice, including consent withdrawal, please see our Privacy Policy .

Reviewer #1: No

Reviewer #2: No

---

## [Author Response · Author response to Decision Letter 1]

25 Feb 2025

Dear Editors and Reviewers:

Thank you for your letter and for the reviewer's comments concerning our manuscript entitled " Social determinants of health on All-Cause and Cause-specific Mortality in US Adults with chronic obstructive pulmonary disease: NHANES 2005–2018.". Those comments are all valuable and very helpful for revising and improving our paper. The main corrections in the paper and the responds to the reviewer's comments are as flowing:

Responses to reviewer 1's comments are provided below:

1. The review's comment: While the statistical analysis is generally sound, several issues warrant attention. The researchers dichotomized the SDoH scores using a cutoff of 2, but the rationale for this specific threshold isn't thoroughly explained. This artificial dichotomization might oversimplify the complex nature of social determinants. Additionally, the study could have benefited from sensitivity analyses using different SDoH score cutpoints to test the robustness of their findings.

The authors' answer: Thank you for your insightful comment regarding the dichotomization of the SDoH scores at a cutoff of 2. The decision to use this threshold was based on prior studies, where a similar dichotomization approach has been commonly applied to differentiate between individuals with lower and higher burdens of unfavorable social determinants of health (SDoH) (References 27, 28). This cutoff provided a practical method for examining the relationship between social determinants and mortality, enabling meaningful comparisons between groups with distinct levels of SDoH.

However, we recognize that dichotomizing complex variables such as SDoH may oversimplify the underlying variability of these determinants, potentially masking important nuances. We appreciate the reviewer’s suggestion to test the robustness of our findings by exploring alternative cutoffs. In response to this, we conducted additional sensitivity analyses using various SDoH score cutpoints. Specifically, we categorized the SDoH scores into tertiles, with Q1 ≤ 2, Q2 ranging from 3 to 4, and Q3 ≥ 5, to examine how the relationship between SDoH and mortality may vary across different levels of social determinants. Importantly, the results remained consistent across these categorizations, reinforcing the robustness of our findings and suggesting that our conclusions hold regardless of the specific cutoff used.

In addition to these analyses, we also treated SDoH scores as a continuous variable within our sensitivity analysis framework. By doing so, we aimed to capture the full range of variation in SDoH, without imposing arbitrary thresholds. This continuous variable analysis demonstrated that the relationship between SDoH and mortality remained stable, further supporting the robustness of our conclusions across different analytical approaches.

Furthermore, to account for potential variations in the effects of SDoH among different patient groups, we conducted subgroup analyses based on factors such as age, sex, ethnicity, BMI, hypertension and CVD. These analyses confirmed that the impact of SDoH on mortality was consistent across various subgroups, providing additional evidence for the stability and generalizability of our findings.

We have now included these additional sensitivity and subgroup analysis results in the revised manuscript. Detailed data from the sensitivity analyses and subgroup analysis results, including both the categorical and continuous SDoH approaches, can be found in Supplementary Table 2 and Figure 4 (A-C). We believe these comprehensive analyses address the reviewer’s concerns and offer a more robust understanding of the relationship between SDoH and mortality in COPD patients.

2. The review's comment: Interpretation and Generalizability: The authors appropriately acknowledge that their findings may be most applicable to the U.S. population, given the NHANES data source. However, they could have provided more discussion about how their results compare to similar studies in other countries or healthcare systems. The linear relationship found between SDoH and mortality outcomes seems somewhat surprising given the complex nature of social determinants, and this finding deserves more critical examination.

The authors' answer: Thank you for your insightful feedback regarding the interpretation and generalizability of our findings. We would like to emphasize that our study was based on a weighted analysis using data from the National Health and Nutrition Examination Survey (NHANES), which is a nationally representative sample of U.S. adults. This methodological approach ensures that our findings reflect the broader U.S. population, making the results more generalizable within this context.

To further strengthen the reliability of our findings, we conducted sensitivity analyses using different SDoH score cutoffs. Specifically, we divided participants into tertiles based on their SDoH scores (Q1 ≤ 2, Q2 3-4, Q3 ≥ 5). These analyses demonstrated that the relationship between a higher SDoH burden and increased mortality risk remained consistent, regardless of the cutoff used. This added layer of analysis underscores the robustness of our conclusions and supports the validity of the observed associations. Furthermore, we also analyzed SDoH as a continuous variable within our sensitivity analysis framework. The results from this continuous variable analysis showed that the relationship between SDoH and mortality remained stable, further reinforcing the robustness of our findings.

In addition to these sensitivity analyses, we conducted stratified analyses based on key variables, such as age, gender, and disease severity, to explore whether the impact of SDoH on mortality differed across patient subgroups. These analyses revealed that the association between SDoH and mortality was consistent across these subgroups, suggesting that the effect of social determinants is stable regardless of individual patient characteristics. This strengthens the generalizability of our findings to various patient populations and further supports the validity of the observed associations.

We also expanded the discussion by comparing our results with similar studies conducted in other countries or healthcare systems. Research from China, Denmark, and other parts of the U.S. has shown similar associations between SDoH and mortality outcomes, although the strength and mechanisms may differ depending on local healthcare systems, social policies, and economic conditions. Studies cited in references 33-36, 19, and 47-51 provide important comparisons, showing consistent findings of adverse social determinants leading to poorer health outcomes, thus reinforcing the broader relevance and generalizability of our results.

Regarding the linear relationship between SDoH and mortality outcomes, we recognize that this might appear surprising, given the complexity of social determinants. However, our use of robust statistical methods, including Cox proportional hazards regression and restricted cubic spline (RCS) analysis, ensures the appropriateness of our findings. The lack of significant nonlinearity in our analysis suggests that in the U.S. context, even relatively small increases in SDoH burden result in a consistent rise in mortality risk. This aligns with findings from similar studies, as referenced in studies 47-49, where cumulative effects of multiple social factors were shown to impact health outcomes in a linear manner. We also acknowledge that while the linear relationship observed in our study is statistically robust, future research may benefit from further exploration of potential interaction effects and nonlinearity in the relationship between specific SDoH factors and mortality.

In summary, our study provides robust evidence of the relationship between SDoH and mortality in COPD patients. The weighted analysis, sensitivity analyses with varying cutoffs and continuous SDoH variable analysis, along with appropriate statistical methods, all contribute to the strength and reliability of our findings. By comparing our results with similar studies in other countries and incorporating subgroup analyses, we further demonstrate the broad relevance of these findings and the importance of addressing SDoH in public health strategies. We hope these revisions sufficiently address your concerns and enhance the clarity and depth of our manuscript.

3. The review's comment: Discussions: A more detailed discussion of the potential mechanisms linking SDoH to mortality outcomes, or analysis of potential interactions between different SDoH components are recommended. For example, consider discussing Physical activity and prevention of mental health complications: an umbrella review. This would provide valuable insights into how physical activity interventions could potentially mitigate some of the adverse effects of poor SDoH in COPD patients, particularly given that the authors described increased mortality risks associated with higher SDoH scores. Also, the effectiveness of community ageing in place, advancing better living for elders as a biobehavioural environmental approach for disability among low-income older adults is also recommended. This article would be relevant for understanding community-based interventions that could help address SDoH factors in vulnerable populations, similar to those identified in your COPD study.

The authors' answer: Thank you for your valuable suggestions regarding the discussion of potential mechanisms linking social determinants of health (SDoH) to mortality outcomes, as well as the inclusion of relevant studies on physical activity and community-based interventions. We appreciate the opportunity to expand on these points in the revised manuscript.

We have added a more detailed discussion on the potential mechanisms through which SDoH may influence mortality outcomes in COPD patients. We acknowledge that the relationship between SDoH and health outcomes is complex and multifaceted, involving a variety of biological, psychological, and social pathways. One key mechanism is the impact of socioeconomic factors on health behaviors, such as physical activity, nutrition, and smoking, which are known to affect COPD outcomes. We have emphasized the importance of addressing lifestyle factors and behaviors, particularly in relation to physical activity, which can mitigate some of the adverse effects of poor SDoH. As you suggested, we have referenced the umbrella review on physical activity and its role in preventing mental health complications, as this provides important insights into how physical activity interventions could help alleviate the health burden associated with low SDoH in COPD patients (References 58-60). Regular physical activity has been shown to improve lung function, reduce symptoms, and enhance mental well-being, particularly in low-income or disadvantaged populations, making it an important intervention for improving health outcomes.

Additionally, we have expanded the discussion to include community-based interventions, particularly focusing on "community ageing in place, advancing better living for elders," which has been shown to be effective in addressing disability and improving health outcomes in low-income older adults (Reference 61). This approach emphasizes the role of the environment, social support, and accessible healthcare in reducing the negative effects of poor SDoH, and is highly relevant to COPD patients, many of whom are older adults facing multiple social and health challenges. Community-based interventions, such as improving access to healthcare, promoting social engagement, and enhancing living conditions, can help address SDoH factors and mitigate their impact on health. We believe that integrating these strategies into COPD management could improve survival rates and quality of life for patients in vulnerable populations.

Incorporating these mechanisms and interventions into the discussion provides a more holistic view of how SDoH contribute to mortality in COPD patients, and how targeted interventions can help reduce these risks. We hope this expanded discussion adds depth to the manuscript and addresses your concern about the need for a more detailed exploration of potential solutions to the challenges posed by SDoH.

We appreciate your thoughtful feedback and believe that these additions enhance the relevance and applicability of our findings to real-world public health efforts.

Responses to reviewer 2's comments are provided below:

1. The review's comment: Introduction: The authors do a suitable presentation of the health determinants framework (although missing key references such as Dahlgren and whitehead work or Barton). Nonetheless, I consider that the text needs to be arranged in a way that the SDoH are presented regarding its impact on COPD, or respiratory mortality, instead of generally. I also recommend authors to include the aim of the text in a clearer way (and more aligned with the redaction present at the abstract)

The authors' answer: Thank you for your insightful comment regarding the presentation of the health determinants framework and the introduction of Social Determinants of Health (SDoH) in relation to COPD and respiratory mortality. We greatly appreciate your suggestion to refine the discussion by focusing more specifically on the impact of SDoH on COPD, rather than presenting them in a more generalized context. Your feedback has helped us clarify and strengthen the introduction to ensure that it is directly aligned with the aim of the study.

In response to your feedback, we have thoroughly revised the introduction to provide a more focused and contextually relevant discussion of how SDoH specifically impact the development, progression, and outcomes of COPD, as well as the risk of respiratory mortality. The revised structure now highlights the critical social determinants, including socioeconomic status, healthcare access, education, and neighborhood environment, and how these factors directly contribute to disparities in COPD outcomes. By shifting the focus from a broad, general discussion of SDoH to a targeted examination of their role in respiratory health, we aim to draw a clearer connection between these determinants and the health of individuals with COPD. This reorganization ensures that the reader is immediately introduced to the relationship between SDoH and COPD, which is central to the objectives of the study.

Furthermore, we have explicitly stated the aim of the study in a clearer and more precise manner, aligning it with the framing and language used in the abstract. In the revised introduction, we now directly state that the primary objective of the study is to evaluate the association between SDoH and the risk of all-cause and cause-specific mortality in COPD patients. This revision enhances the transparency of the study’s goal, making it evident from the outset that the focus is on understanding the role of social determinants in influencing mortality outcomes for COPD patients, a theme that is consistently addressed throughout the manuscript.

We also recognize the importance of grounding our study within the broader theoretical context of health inequalities. As per your suggestion, we have incorporated key references, such as the seminal work of Dahlgren and Whitehead, and Barton’s contributions to understanding social determinants and health inequalities. These references offer important background for understanding how social factors shape health outcomes at both the individual and population levels, and they provide essential context for interpreting the findings of our study. By situating our research within these well-established frameworks, we further reinforce the significance of addressing SDoH in public health strategies and policy development.

Overall, we believe that these revisions substantially improve the clarity and focus of the introduction. By reorganizing the content to emphasize the direct impact of SDoH on COPD and respiratory mortality, and by aligning the aim of the study with the abstract, we have created a more coherent narrative that clearly defines the scope and objectives of the research. Additionally, by incorporating the suggested references, we have strengthened the the

---

## [Decision Letter · Decision Letter 1]

25 Mar 2025

Social determinants of health on All-Cause and Cause-specific Mortality in US Adults with chronic obstructive pulmonary disease: NHANES 2005–2018.

PONE-D-24-59243R1

Dear Dr. Tu,

We’re pleased to inform you that your manuscript has been judged scientifically suitable for publication and will be formally accepted for publication once it meets all outstanding technical requirements.

Kind regards,

Angela Mendes Freitas

Academic Editor

PLOS ONE

Additional Editor Comments (optional):

Reviewers' comments:

Reviewer's Responses to Questions

**Comments to the Author**

1. If the authors have adequately addressed your comments raised in a previous round of review and you feel that this manuscript is now acceptable for publication, you may indicate that here to bypass the “Comments to the Author” section, enter your conflict of interest statement in the “Confidential to Editor” section, and submit your "Accept" recommendation.

Reviewer #1: All comments have been addressed

Reviewer #2: All comments have been addressed

2. Is the manuscript technically sound, and do the data support the conclusions?

Reviewer #1: Yes

Reviewer #2: Yes

3. Has the statistical analysis been performed appropriately and rigorously? 

Reviewer #1: Yes

Reviewer #2: Yes

4. Have the authors made all data underlying the findings in their manuscript fully available?

Reviewer #1: Yes

Reviewer #2: Yes

5. Is the manuscript presented in an intelligible fashion and written in standard English?

Reviewer #1: Yes

Reviewer #2: Yes

6. Review Comments to the Author

Reviewer #1: All comments have been thoroughly addressed. I extend my gratitude to both the authors and editors for taking my opinions into consideration during the review of this manuscript.

Reviewer #2: The authors addressed all the issues raised and the text was significantly improved. It is a valid contribution to the discussion on the association between health determinants and health outcomes.

7. PLOS authors have the option to publish the peer review history of their article (what does this mean? ). If published, this will include your full peer review and any attached files.

**Do you want your identity to be public for this peer review?** For information about this choice, including consent withdrawal, please see our Privacy Policy .

Reviewer #1: No

Reviewer #2: No

---

## [Editor Report · Acceptance letter]

PONE-D-24-59243R1

PLOS ONE

Dear Dr. Tu,

I'm pleased to inform you that your manuscript has been deemed suitable for publication in PLOS ONE. Congratulations! Your manuscript is now being handed over to our production team.

Kind regards,

on behalf of

Dr. Angela Mendes Freitas

Academic Editor

PLOS ONE